# Mechano-Transduction Boosts the Aging Effects in Human Erythrocytes Submitted to Mechanical Stimulation

**DOI:** 10.3390/ijms231710180

**Published:** 2022-09-05

**Authors:** Simone Dinarelli, Giovanni Longo, Antonio Francioso, Luciana Mosca, Marco Girasole

**Affiliations:** 1Institute for the Structure of the Matter (ISM), Italian National Research Council (CNR), Via Fosso del Cavaliere 100, 00133 Roma, Italy; 2Department Biochemical Sciences “A. Rossi-Fanelli”, Sapienza University of Rome, Piazzale A. Moro 5, 00185 Roma, Italy

**Keywords:** erythrocytes, cell aging, mechanical sensing, mechano-transduction, AFM, membrane roughness, metabolic adaptation

## Abstract

Erythrocytes’ aging and mechano-transduction are fundamental cellular pathways that determine the red blood cells’ (RBCs) behavior and function. The aging pattern can be influenced, in morphological, biochemical, and metabolic terms by the environmental conditions. In this paper, we studied the effect of a moderate mechanical stimulation applied through external shaking during the RBCs aging and revealed a strong acceleration of the aging pattern induced by such stimulation. Moreover, we evaluated the behavior of the main cellular effectors and resources in the presence of drugs (diamide) or of specific inhibitors of the mechano-transduction (probenecid, carbenoxolone, and glibenclamide). This approach provided the first evidence of a direct cross-correlation between aging and mechano-transduction and permitted an evaluation of the overall metabolic regulation and of the insurgence of specific morphological features, such as micro-vesicles and roughness alterations. Overall, for the first time the present data provided a schematic to understand the integration of distinct complex patterns in a comprehensive view of the cell and of its interactions with the environment. Mechano-transduction produces structural effects that are correlated with the stimulation and the strength of the environmental stimulation is paramount to effectively activate and trigger the biological cascades initiated by the mechano-sensing.

## 1. Introduction

Erythrocytes (RBCs) are among the most studied cells in our body. Their physiological role is to transport O_2_ and CO_2_ in the blood; a task that requires the reversible binding of these molecules to the hemoglobin (Hb) for which the RBCs constitute a safe and functional envelope.

There are many peculiar properties of these cells, which make them an interesting subject of study from the biological, clinical, and heuristic points of view. Among the most intriguing characteristics of erythrocytes, we recall: (i)their exceptional mechanical properties, dependent on the structure and composition of the cell skeleton, which ensures protection to the cellular microenvironment and guarantees a reversible deformability that allows the cell to squeeze through the narrow capillaries and perform properly its biological function.(ii)their relatively simple cellular biochemical machinery, which determines a metabolism essentially relying on glycolysis, for the production of energy (ATP), and on the pentose phosphate (PPP) shunt for the production of reducing power (NADPH).(iii)the role of transmembrane pumps, mostly ATP dependent, that ensure ionic balance, but also act as a physical support for some fundamental enzymatic activities.(iv)the absence of a nucleus and of nucleic acids, which makes the RBCs unable to self-replicate, causing the blood homeostasis to rely on a dynamic balance between newborn cells produced in the bone marrow and aged cells removed from the circulation in kidneys and spleen.

These observations introduce two important aspects of the present paper. On one hand, the importance of cellular aging, which determines the duration of cellular life and, on the other hand, the special relationship of RBCs with the environment that influences their morphology and modulates their function [1]. Here, we will focus on a class of RBC-to-environment interactions of particular importance, that is the ability of erythrocytes to detect mechanical signals from the environment and to process them into biochemical signals and metabolic patterns. This is mostly due to an efficient mechano-sensing and mechano-transduction system that consists of a complex and still not completely understood cascade of events initiated by mechanical deformations sensed by the transmembrane PIEZO 1 protein. The mechano-sensing activates downstream events that, mediated by calcium fluxes and metabolic regulations, results in the expulsion of intracellular ATP through Pannexin 1 (PANX 1) and Cystic Fibrosis Transmembrane Regulator (CFTR), as well as in cellular dehydration [2,3,4,5].

Interactions with the environment are fundamental in the cellular aging process and in the past years our group has studied such phenomenon under different laboratory conditions and has developed and refined a very robust and sensitive experimental model capable of highlighting the effects of chemical agents, pathologic effector (such as beta amyloid), or environmental stimuli [6,7,8]. In particular, among these we studied the effects of simulated microgravity on cell aging [9], which provided an interesting base of comparison for the biochemical, metabolic, and morphologic adaptation that occurs in RBCs in the presence of mechanically altered environmental conditions.

Following previous studies [9,10], here we employed a model of artificial aging for RBCs’, which was particularly interesting for the study of the mechanical stimulation: For instance, the incubation of cells in the absence of glucose and extracellular calcium, which allowed to accelerate the “in vitro” aging by hampering the reload of energy resources. Furthermore, the absence of extracellular calcium substantially eliminated the events mediated by this second messenger, thus highlighting its physiological role and providing the opportunity to evaluate secondary mechanisms for the processing of the mechanical stress. In this way, our experimental framework highlighted aspects of cellular function and metabolism under stress, including the events occurring in extremely low ATP conditions (around 1/1000 of the normal value) and allowed understanding the role of mechanical stimulation in such extreme conditions. 

Overall, the purpose of this work is to understand the influence and the relative importance of the processing of the mechanical stimulation in the regulation of the aging dynamics of human RBCs. In particular, it is interesting to shed some light on the involvement of mechano-transduction pathway in the regulation of the aging progression pathway, which can be speculated on the base of common morphological and biochemical mediators occurring in these two pathways. An example of common intermediate between aging and mechano-transduction is provided, in morphological terms, by the increase of spherocytes, which is well known to occur in both cases as well as in pathological states associated to mechano-transduction impairment [6,11,12]. 

To this end, we have compared the aging pathway of RBCs kept in static conditions with cells exposed to a relatively modest mechanical stimulation, i.e., a continuous shaking performed with a tilting shaker. The involvement and relative importance of the mechano-transductive cascades has been investigated by comparing the pathways observed in static or mechanically-stimulated RBCs in the presence and absence of chemical agents that reduce the sensing of the environmental stimulus or inhibits molecular channels, such as PANX 1 and the CFTR, that are involved in the later steps of the mechano-transduction. To this end, we employed the drugs diamide (DIAM), probenecid (PROB), carbenoxolone (CARB), and glibenclamide (GLIB) [4,13,14].

This study employs an approach based on the combination of conventional biochemical techniques with qualitative (optical) and quantitative (Atomic Force Microscopy, AFM) microscopy. While the first techniques allow for monitoring the energy resources and the main functional indicators of the cells, the latter provides an assessment of the cell morphology with sensitivity on the nanoscale and highlights the occurrence of different morphological intermediates in the aging pathways and in different experimental conditions. In this context, AFM-based approaches have also been used to evaluate the structural integrity of the cell membrane-skeleton through an analysis of the membrane arrangement (i.e., analysis of the membrane roughness) providing a valuable quantitative method to understand the structural dynamics related to the RBCs aging [15,16].

## 2. Results

The main objective of the present work is to study the effects of an external mechanical stimulation on the aging path of RBCs; therefore, it is essential to carefully define this mechanical stimulation. We have chosen a relatively modest stimulation corresponding to a continuous shaking delivered through a tilting shaker (with an angle of about 15 degrees), such as those typically used for mixing effectors added to samples or to keep cells in suspension. In mechanical terms, this stimulation corresponds to a continuous (non-vorticose) shaking of the cells, in which the mechanical effect is substantially attributable to collisions between erythrocytes and, therefore, the expected effects will depend on the density of the fluid and the number of cells per unit of volume (i.e., hematocrit, which in our case was set at 20%). 

The cellular stress conditions, on the other hand, were analyzed by monitoring cellular parameters that were critical for understanding the cellular metabolic condition. In accordance with the methodology followed in previous works [9] we monitored: (i) cell lysis; (ii) the oxidation state of Hb; (iii) intracellular and extracellular ATP; (iv) cellular reducing power GSH/GSSG; (v) and the capacity of the cells to reload their resources after incubation with a revitalizing solution (i.e., IPP solution).

As a whole, these parameters allowed for us to monitor the overall cell conditions, to estimate the oxidative stresses suffered by the cells, and to compare them with the cellular resources in terms of reducing power and residual ATP in different times of the aging process.

As shown in Figure 1, the mechanical stimulus continuously applied during the aging produces an increase in the overall stress suffered by the cells. This enhancement becomes measurable after 4 days of incubation and results in a significant increase in cell lysis compared to the static control. Furthermore, the evaluation of the oxidation rate of Hb (monitored through the measurements of the Soret peak) indicates that the cells, as the time increases, respond less effectively to the oxidizing environment. Indeed, as shown in Figure 1b, after a transient phase of about 3–4 days, the oxidation rate of Hb in mechanically stimulated cells is significantly higher.

To understand whether this effect on the Hb is due to a different use of the cell reducing power, we measured the GHS to GSSG ratio, as well as the intracellular ATP content, at various steps of the aging path for the static control and for the mechanically stimulated cells.

The results are shown in Figure 2 for both the monitored cellular resources. The assessment of cellular reducing power, shown in Figure 2a, provides a clear picture of the phenomenon. 

The first clear evidence is that the shaking environment requires a much higher use of GSH compared to the static controls. As a consequence, the consumption kinetics for the shaken sample is much faster. Furthermore, since for a ratio GSH/GSSG is smaller than one, the reducing power is insufficient to tackle the oxidative stresses, in the shaking condition the cell’s reducing systems stop protecting the cell from the oxidative environment earlier than in static controls (4 or less days instead of 8 or more days). Interestingly, these data are in good agreement with the trend observed for the Hb oxidation data.

As is well known, the balance of reducing power in RBCs is interconnected with the production of ATP through the existing crossover between glycolysis and pentose phosphate pathway (PPP). Therefore, it is important to measure the kinetics of ATP consumption, which, in our experimental conditions (i.e., buffer deprived of nutrients), cannot be restored. 

A typical measured trend is shown in the Figure 2b, which clearly shows an accelerated consumption kinetics for mechanically stimulated samples. This is particularly true in a specific critical phase of the aging, which occurs between the fourth and eighth day of aging. The existence of a critical phase, associated with a higher ATP consumption, was already known from previous studies [9,15]; however, in the case of mechanically stimulated RBCs, the onset of the critical phase is anticipated and is associated with a particularly high consumption of cellular resources even when compared to the controls.

Overall, the reported data shows that the applied mechanical stimulation acts as a powerful accelerator of the aging pathway.

Since the behavior of the metabolic machine in different environmental conditions may play a role in the results we are showing, it seems interesting to probe the effectiveness of the metabolic apparatus during the aging. A simple, effective, and well-established method consists in submitting the erythrocytes to a rejuvenation process by providing the cells with the elementary constituents (inosine, pyruvate, and phosphate) that allows for the metabolically efficient RBCs to restart the production of ATP and reducing power. Clearly, only metabolically active cells can respond to such treatment, while the others will be essentially inert.

The rejuvenation process, performed at different selected aging times (for the data of Figure 3, respectively, after 2, 4, 7, and 10 days), highlights how the metabolic reloading machine loses its effectiveness over time. The loss of efficacy is better assessed by observing its effects on the reloading of ATP (Figure 3) and on the Hb (Figure 4), rather than on the reducing power (that shows smaller effects).

The trend of the ATP reloading as a function of the aging time reported in Figure 3, shows that at the smallest time the ATP recharging is greater in the mechanically stimulated sample. Indeed, at that time, all the cells are metabolically active and the greater reloading is essentially a consequence of the greater consumption of ATP in the mechanically stimulated cells. However, as the aging time increases, compared to the static controls, the metabolic machine of the mechanically stimulated cells loses its effectiveness more rapidly, to the point that it stops reloading completely in less than 7 days. On the contrary, simultaneously, the static samples still shows a residual reloading activity that permits the cells to carry out at least the most urgent needs. 

Finally, in Figure 4 we describe the effects of the revitalization on the intracellular Hb, for both samples. In particular, we followed the re-oxidation kinetics of intracellular Hb that occurs after rejuvenation performed after 2, 4, 8, and 11 days. 

The results, shown in Figure 4 for the revitalization performed after 2 days (Rej 1), 8 days (Rej 3) and 11 days (Rej 4), evidence that the re-oxidation kinetics of Hb, in the mechanically stimulated sample, is faster than in the static samples at every revitalization time.

To complete the investigation on the effects of the mechanical stimulation on the aging pattern of human RBCs, we performed a series of morphological and ultra-morphological analyses with the aim of providing a more in-depth description of the changes induced by the stimulus. These analyses were conducted combining conventional optical microscopy imaging with high-resolution AFM microscopy characterization.

At first, we used standard optical imaging to obtain a preliminary evaluation of the cell morphologies observed in the samples. The cells were divided in four basic morphologies: biconcave (BIC), crenated (CREN), spiculed (SPIC), and spherocytic (SPHE). The results, summarized in Figure 5a,b, describe the relative population of the abnormal morphologies (namely CREN, SPIC, and SPHE), at three aging times and in static and mechanically stimulated samples. 

The resulting morphological patterns indicate that the aging in static and in mechanically stimulated samples leads to qualitatively similar evolutions. However, as the aging time increases, the mechanical stimulation induces a considerably greater presence of spherocytic cells, compared to the static controls. The effect is already visible after five days but grows significantly over the next three days as the intracellular resources decrease.

This morphological information can be matched with the behavior of the intracellular volume that can be measured from the AFM images of the RBCs (Figure 5c). 

Indeed, the RBCs’ aging is typically associated to a decrease in cell volume and as shown in Figure 5, in both the investigated experimental conditions, a reduction in volume is observed over time, but with both quantitative and qualitative differences. In fact, in mechanically stimulated samples, the decreasing trend is monotonous, faster, and quantitatively more significant than in static controls. In particular, in the static control there is an initial transient phase that lasts for at least three days, during which the cell volume does not change. 

The membrane roughness is a morphometric parameter that has been used to monitor the relationship between membrane and the underlying skeleton as it has been shown that an (even reversible) detachment from the membrane results in a significant decrease of the measured roughness values [15,16]. 

In the present case, the roughness trend is shown in Figure 6 for the static controls and the shaken samples and indicates that the aging trend in the presence of mechanical stimulation has a much faster decrease dynamic, which is diagnostic of a faster and larger disengagement of the cell skeleton from the membrane. At longer aging times, in the mechanically-stimulated sample, a slight increase in the measured roughness occurs. This phenomenon likely depends on ruffling and rearrangement of the plasma membrane and is favored by the loose bounding of the membrane to the skeleton that takes place at the long aging time.

To complete the morphological analysis of the aging in static and shaken RBCs, we performed an AFM, imaging-based, investigation. In Figure 7, we show the progression of the aging pathway in the static and mechanically-stimulated cells. In agreement with the data from optical microscopy, the evolution of the cell shapes is qualitatively similar, although in the shaken samples the number and severity of unusual morphologies grows. This is especially true for the intermediate aging times, while at the later aging steps, the presence of swollen or proto-spiculed cells is equally abundant in both samples.

An analysis of the evolution of the plasma membrane is reported in Figure 8, which exploits the nanoscale sensitivity of the AFM imaging to clarify the aging progression.

Comparing the two samples, it can be observed that, while after the first day the two surfaces are almost equivalent, as the time increases, the development of aging markers at the membrane level occurs sooner in the case of the shaken samples compared to the static ones. A typical example is the development of small spicules and proto-spicules on the cells, as well as the development of small vesicles from the cells’ membrane. In particular, micro-vesicles can be observed even at short aging times (e.g., at the third day) in mechanically stimulated samples, while they are barely visible, and only in a very preliminary phase, in the static controls (see the black arrows in Figure 8b). Furthermore, this phenomenon grows significantly after six days, especially in mechanically stimulated cells.

These differences in vesicle production and release can possibly contribute to the difference in the cell volume decreasing, which was observed in these two samples and shown in Figure 5c.

Summarizing, the effects of the mechanical stimulation produce more severe effects on all the morphological parameters we used to evaluate the RBCs aging.

To investigate the relationship between aging and mechanical stimulation, and to understand whether there may be an involvement of the mechano-transduction cascade in the observed accelerated aging for the mechanically stimulated samples, we repeated the experiments in the presence of drugs and inhibitors acting on the mechano-sensing and mechano-transduction pathways.

As first effector, we chose the drug DIAM, that determines a physical stiffening of the plasma membrane, reducing its deformability [4] and generically hindering the mechanical sensing. Moreover, we investigated the effects of specific inhibitors that act on PANX 1 (PROB and CARB) or on CFTR (GLIB) that are the two channels involved in the later steps of the mechano-transduction [13,14].

In Figure 9, the effect of the employed chemicals and inhibitors on the percentage of cell lysis is shown. As previously noted, the mechanical stimulation causes an increase of cell lysis. As shown in Figure 9a,b, this increase is a general phenomenon that occurs in all the samples, in the presence or absence of inhibitors. The data on the cell lysis in presence of PROB, GLIB, and CARB show that the drugs protect the cells from lysis in the given experimental condition (i.e., static drug vs. static control or, shaken drug vs. shaken control), while no difference in this sense can be seen in the presence of DIAM. 

Figure 9c,d show the comparison between the lysis ratios (CTR/PROB and CTR/DIAM) in static and mechanically stimulated conditions. In the presence or absence of the mechanical stimulation, the two trends remain within or at the limit of the experimental error. This means that, while the inhibitors can actually protect cells from lysis, their protective effect does not change in the presence and absence of mechanical stimulation, thus the protection is not related to the presence of a mechanical-stimulus and, as consequence, inhibition of the mechano-transduction does not prevent cell lysis. 

In other words, the mechano-transduction does not affect the cell lysis. 

On the contrary, all the chosen drugs have an important effect on the metabolism of ATP. Indeed, as shown in Figure 10, during the critical aging phase associated with a qualitative change in cellular behavior and with a higher energy expenditure (4–7 days), the consumption of ATP in the mechanically-stimulated cells in presence of the drugs is significantly reduced with respect to the static controls. This data provides clear evidence that the protection/regulation of metabolism is uniquely and indissolubly associated with the presence of the mechanical stimulation.

Interestingly, depending on the molecular mechanism of the drug actions, the ATP saving can be observed at all the aging times (DIAM) or just concomitant to critical phases (GLIB and PROB).

An interesting aspect is that, despite the significant effect observed on intracellular ATP metabolism, PROB, and GLIB have only a modest and rather non-specific effect on the GSH/GSSG ratio. Similarly, the measured extracellular ATP in static controls and in mechanically stimulated samples were not significantly different.

Overall, the reported data clearly indicate a cross-correlation between aging and mechano-transduction, since (partial) inhibitors and chemicals interfering at various levels with the mechano-transduction reduce the aging acceleration induced by the mechanical stimulus, but play no role in its absence (i.e., in static conditions). 

## 3. Discussion

One of the main purposes of this study consists in the search for a correlation between the erythrocytes’ aging and the cell mechano-transduction. 

The occurrence of common morphological intermediates in both these fundamental patterns, for instance the presence of a high spherocytic rate, suggests that some biophysical or biochemical upstream effector could exist and could act as a common regulator, in such a way that the two phenomena could be mutually influenced. In this sense, it is worth noting that the cell morphology in RBCs is much more informative of the cell status compared to other kinds of cells; in erythrocytes the shape results from a variety of interactions with the environment, modulated by the internal biochemical pathway and the metabolic balancing.

The conservation of the RBC’s shape is considered of strategic importance to guarantee an optimal gas exchange, as it allows exposing the maximum surface area through which O_2_ and CO_2_ can freely diffuse [17]. Moreover, the cell shape and volume also contribute to the rheological properties of the erythrocytes that are necessary to ensure their proper biological function in the circulatory system. For this reason, the cell morphology and volume are controlled, by the membrane-skeleton in a very accurate way, at least as long as the cellular resources permit such control. Following the aging pattern of RBCs through morphological and biochemical techniques allows to understand in detail what happens when the RBCs are running out of energy and of reducing power and what kind of distinctive pattern of aging can be triggered as function of the environmental conditions.

As a first step in the investigation of the relationship between aging and mechano-transduction, we analysed the effect of mechanical stimulation on the characteristics of the aging pattern of human RBCs. 

In this framework, all the evidence collected in this study point toward the fact that the mechanical stimulation delivered during the cell incubation induces a strong acceleration of the aging pathway. Indeed, the mechanical stimulation produces a higher rate of cell lysis, a faster intracellular hemoglobin oxidation, a remarkably higher consumption of ATP and of reducing power and a faster impairment of the metabolic machine as long as the reload of energetic resources is involved (i.e., rejuvenation experiments). 

Some of these parameters are correlated, since in RBCs the balancing of ATP and NADPH is under metabolic control and, the synthesis of these two important molecules can be shunted through the PPP in order to produce one or the other according to the most urgent cell needs. As a consequence, the cellular response can be adapted to the environmentally-induced requests. For instance, most of the reducing power is used for the reduction of metHb (by Cyt b5 or Flavine reductase) and to tackle the oxidative stresses (GSH). The reducing agents are NADH, produced in the last step of glycolysis and NADPH, (used for GSH and flavine reductase) which is produced in the PPP. In practice the cell can privilege either energy production and met-Hb control through the glycolysis or production of GSH and Hb control in the PPP. Theoretical models suggest that the metabolic shift between these two pathways is governed by the Hb oxidation state and rates [18].

Equally important are the differences observed in morphological terms in the two samples and that consist in an (expected) enhancement of spherocytic morphologies, in a faster cell volume decrease and in an accelerated occurrence of morphological markers of aging (e.g., micro-vesicles, proto-spicules etc.) in mechanically-stimulated cells. All these data fit in a landscape that, in full agreement with the biochemical data, is clearly indicating an acceleration of the aging pathway induced by the mechanical stimulation. 

In agreement with this statement is also the evaluation of the decreasing trend of the membrane roughness that, in the mechanically stimulated cells, is diagnostic for a less supported membrane structure related to a more pronounced detachment of the cell skeleton from the overlying membrane. [6,15,16]

Clearly, the biochemical and morphological data are associated and interconnected. In this sense, for instance, it is interesting to remind that previous studies on the RBCs’ aging suggested that the shortage of ATP has a major effect on the cell structure and morphology, while the lack of reducing power produces more clear effects on the Hb. 

Indeed, the stability of the RBCs’ skeletal proteins is correlated to their phosphorylation status, which is controlled by protein kinase C (PKC). A shortage of ATP activates PKC and can result in a transient dissociation of membrane from the cell skeleton [19,20,21]. These morphological alterations can evolve in condition of severe stresses. For instance, the caspase-3 cascade can be activated by PKC and shortage of ATP, causing the development of stable structural defects to the cell skeleton [8,22,23]. 

In the light of these considerations, the interpretation of the data presented here is straightforward: the continuous shaking of the cells produces, through a variety of mechanisms, a disturbance of their structure and stability [24] that can be counteracted by the RBCs only through an acceleration of their metabolism that results in a faster consumption of ATP and reducing power. After the first few days, the higher demand produces a progressively more severe shortage of cell resources (GSH already after the third day and ATP in the next days) that, possibly through the involvement of PKC or caspases, is translated into all the biochemical and morphological evidence here reported (e.g., Fe-Hb less reduced, membrane structural weakness, production of micro-vesicles etc.).

In the second part of the study, we tried to understand whether the observed acceleration of cell aging effects could be related to molecular determinants of the mechano-transduction. To this end, we followed the RBCs’ aging in the presence of inhibitors and drugs affecting the mechano-transduction and collected clear evidence that, despite no alteration in cell lysis rate can be measured, a significant reduction in the ATP consumption takes place only in the presence of mechanical stimulation. 

Interestingly, since the molecular mechanism used by the drugs to hamper the mechano-transduction are different, the effects on the cells’ energetic metabolism varies correspondingly. For instance, in the case of DIAM, the effect is a pure membrane stiffening, which results in a simple reduction of the mechanical solicitation and, as consequence, the biochemical effect can be observed at all aging times. Alternatively, when the drug acts through a regulation/inhibition of a mechano-transduction factor, this effect is somehow under a metabolic control. Namely, their effect is little during the first days when the cells’ resources are slowly consumed, but their effect becomes relevant during a critical aging phase (i.e., approximately between the fourth and seventh day) when the consumption of resources increases dramatically. Remarkably, in the case of PROB and CARB, which have the same molecular target, the effect measured on the metabolism of ATP is identical (see Appendix A for CARB).

In this framework, also the higher presence of spherocytic cells in mechanically stimulated samples agrees with a role for the mechano-transductive channels as it has been observed in previous studies [25].

The simplest explanation for the whole data is that in stressing conditions an internal metabolic rearrangement takes place in order to privilege reducing power against ATP. On the other hand, we must consider that the mechano-transduction has multiple pathways bound to signal sensing and to the control of upstream and downstream events of the cascade. Yet, in our experimental conditions, we only blocked one way at a time; therefore, we did not expect a complete blockade of the entire mechano-transduction but just its modulation. In this view, we cannot exclude that a partial processing of the mechanical stimulus can still occur in the cells. In this sense, the fact that the main target of the inhibitor-induced protection is the ATP, which is a major determinant of the aging, but that only little effects have been observed on the reducing power could be related to a priority consumption of the reducing power associated to the mechano-transduction phenomenon that is partially still active.

At this stage of the discussion, a comparison between different intensity of the stimulation and between different kinds of mechanical stimulation can be useful in order to understand what kind of new events and phenomena can be triggered by a larger mechanical stimulation. For instance, we already cited the case of diamide that, by stiffening the membrane, produces a quantitative reduction of the mechanical signal that results in a measurable reduction of the biochemical effects, but much a smaller stimulation can be considered. Indeed, in a previous study we investigated the RBCs’ aging in artificial microgravity conditions [9], an environment that provides the cells with a continuous mechanical stimulation of low intensity, which is intermediate between the shaking of the present study and the static controls. Indeed, in our previous study [9], we concluded that the microgravity itself is felt by the cells as a (very light) mechanical stimulation, which is capable to activate a few biological responses that are compatible with the existence of an active mechano-transduction machine. Examples are the expulsion of ATP and acceleration of the roughness decreasing, although some additional, mild metabolic rearrangement takes place as well. 

In the present case, due to the larger magnitude of the stimulus a fairly different biological scenario takes place. The effects are dominated by the large consumption of internal resources that the cells must use in order tackle the stress induced by the environmental conditions. Such high depletion of resources requires a fast metabolic adaptation and is probably superimposed to the activation of biochemical cascades, leading to significant structural consequences, for instance resulting in a much faster decreasing trend of the membrane roughness compared to the static controls and to the microgravity condition. 

More generally, concerning the kinetics of the membrane roughness, the contribution of the mechano-transduction produces structural effects that are correlated to the stimulation. This contribution is measurable and capable to induce a significant effect on the structural components (membrane-skeleton) of the RBCs. In the long term, following the predicted metabolic response subsequent to activation of PKC and, possibly, caspase cascades, the structural alterations are fixed in distinct morphological patterns, depending on the environmental stimulation. The behavior of the roughness at longer aging times is also interesting. Indeed, the small increase of roughness value, possibly resulting from membrane ruffling, can take place only when the structural support of the skeleton is compromised and the membrane is severely stressed, and in the present experiment occur only in the mechanically stimulated samples.

Interestingly, the present data confirm that, in the context of the aging pathway of RBCs, a quantitative ultrastructural analysis represents a very interesting approach as it has been shown that the measured morphological alterations actually recapitulate the differences in the mechanical stimulation experienced by the cells and their metabolic consequences. 

In this sense, a result of this study, especially if placed in comparison with other similar works using different mechanical alterations, is that the strength of the environmental stimulation is extremely important to effectively activate and trigger the biological cascades initiated by the mechano-sensing.

Remarkably, it is worth stressing that we showed the occurrence of biochemical effects certainly induced by mechano-transduction and correlated to a sharp acceleration of the aging pathway in human RBCs in the total absence of extracellular calcium, which is thought, in general terms, to be an enhancer or mediator of the mechano-transduction. It is clear, thus, that the mechanical signal, can be processed by the cells through different pathways with the common result of regulating the RBCs metabolism under stress conditions.

## 4. Materials and Methods

### 4.1. Sample Preparation

Blood samples were obtained through venipuncture into Vacutainers (Becton-Dickinson, Franklin Lakes, NJ, USA) were immediately diluted 2-fold in the buffer solution (10 mM sodium phosphate, 140 mM NaCl, EDTA 1 mM as anticoagulant and adjusted to pH 7.4 with NaOH), then centrifuged for 10 min at 3000 rpm at 4 °C. The platelet-rich supernatant and the white layer on top of the pellet (i.e., leukocytes) were discarded, while an aliquot of the supernatant plasma was stored at 4 °C for the AFM smears. After 4 cycles of re-suspension and washing in the same phosphate buffer (it should be noted that the selected buffer is calcium-free and glucose-free) phosphate, the isolated RBCs were stored under sterile conditions, at room temperature (20 ± 1 °C) at 20% of volume fraction in the buffer solution and then split into aliquots to deliver the chemical treatments. To all RBC’s solutions a protease inhibitor (phenyl-methyl-sulfonyl-fluoride, 1 mM) was added in order to avoid proteolytic degradation along the aging. Two different control samples (untreated RBC’s, addressed as control and dimethyl-sulfoxide DMSO at 0.1%) were needed to fulfil the requirements of the experiment since three chemicals (GLIB, PROB, and CARB) must be diluted in DMSO to be delivered to the cells, while the DIAM is soluble in water. The concentrations of the drugs stock solutions have been prepared in such a way to obtain a final concentration of 0.2% DMSO for all the samples while the final concentrations of the drugs were: 1 mM for PROB, 400 uM for DIAM, 20 μM for CARB, and 20 μM for GLIB. These concentrations were calibrated according to their expected effects, using methods from previous studies available in literature. The working concentration of DMSO (i.e., 0.1%) was selected after a dedicated experimental run, evaluating concentrations from 0.02% up to 2%, in order to obtain the maximum possible DMSO content that grants a null effect on the percentage of lysis with respect to the untreated sample.

### 4.2. Mechanical Stimulation

The RBC samples exposed to the shaking conditions were aliquoted in 2 mL Eppendorf vials. The samples were continuously shaken through a tilting shaker, setting the oscillation angle to 15 degrees and one complete oscillation every 2 s. 

The static samples were maintained in vertical position and gently stirred twice per day in order to avoid hypoxia in the vials.

### 4.3. Spectrometer Measurements (Percentage of Lysis and Hb Oxidation State)

To evaluate the progression of the aging phenomena under the different mechanical and chemical treatments, we monitored the percentage of cells that lysed over time and the oxidation state of the haemoglobin (Hb), both intracellular and extracellular, by means of a spectrophotometric method. The spectrum of each sample was acquired daily through a double-beam spectrophotometer Jasco V-630, in the range 350–700 nm, where the three characteristic peaks of the Hb appear. The prominent peak, around 414 nm (the Soret peak), has been used to evaluate the parameters according to the following procedure: (i)for the extracellular Hb, 60 μL of the cellular solution were taken and diluted 1:20 in buffer solution, then centrifuged at 3200 rpm for 12 min in order to remove the intact cells from the solution. An aliquot of 800 μL of the supernatant was taken and diluted 1:1 in buffer and then measured. The wavelength of the Soret Peak has been used to evaluate the oxidation state of the extracellular Hb.(ii)for the intracellular Hb, 6 μL of the cellular solution was diluted 1:200 in bi-distilled water to completely lyse the cells. Next, the sample was centrifuged at 3200 rpm for 12 min and 1 mL of the supernatant was further centrifuged at 11,000 rpm for 11 min to remove the cellular debris from the solution. An aliquot of 800 μL of the supernatant was taken and diluted 1:1 in bi-distilled water and immediately measured. The wavelength of the Soret Peak has been used to evaluate the oxidation state of the intracellular Hb. (iii)from the above-mentioned spectra, the percentage of lysis has been calculated from the ratio between the values of absorbance of the extracellular vs. intracellular (i.e., 100% lysis) spectra. 

It is worth noting that we evaluated during of the aging, the influence of errors due to the pipetting of the aliquots and we have considered them negligible. Moreover, our experimental procedures ensured that the measured aliquots were free of other non-RBCs, bilirubin and plasma. Thus, reasons we did not correct the spectral baselines prior to the data extraction.

### 4.4. ATP

The measurements of the intracellular ATP levels were performed every 2–3 days, by means of the kit “Cell Titer-Glo Luminescent Call Viability Assay” (Promega) and a Wallac 1420 VICTOR3 V plate reader. The procedure to prepare and measure each sample was the following: (i)a 200 μL aliquot of the sample was diluted 1:1 in buffer solution and then centrifuged at 3200 rpm for 12 min, then all the supernatant was carefully removed and the remaining pellet was diluted 1:5 (*v*/*v* ratio) in perchloric acid 0.6 M; gently mixed and then centrifuged at 8000 rpm for 8 min, in order to remove all the denatured proteins from the sample. An 80 μL aliquot of the supernatant was neutralized during 1 h of incubation in an ice bath in presence of 6:1 potassium carbonate 2.5 M.(ii)A 240 μL aliquot of the supernatant were centrifuged at 11000 rpm for 11 min in order to eliminate any residual pellet and an aliquot of 200 μL of this supernatant was used for the measurement of the intracellular ATP. (iii)For each sample two aliquots of 65 μL each were measured in two different 96 multiwall plates, by adding 65 μL of the reagent and, according to the recommendations of the manufacturer, incubate the solution for 10 min in a dark environment.

For the intracellular ATP: the pellet was diluted 1:5 (*v*/*v* ratio) in perchloric acid 0.6 M; gently mixed and then centrifuged at 8000 rpm for 8 min to remove all the denatured proteins from the sample. A fraction of the supernatant (typically 80 μL) was neutralized in a new vial, by adding 6:1 (*v*/*v* ratio) potassium carbonate 2.5 M followed by 1 h incubation in ice bath. Next, 240 μL of the supernatant were centrifuged at 11000 rpm for 11 min to eliminate any residual pellet. A 200 μL fraction of this supernatant was used for the measurement of the intracellular ATP.

All the measurements were performed after 20, 30, and 40 min from the mixing with the reagent. To correlate the counts with the concentration of ATP, each measured plate contained a standard calibration series of ATP that were freshly prepared before the measurements by progressive dilution of a batch 16.5 M ATP solution (Sigma-Aldrich, St Louis, MO, USA). The obtained results were corrected according to the percentage of cell lysis to refer the values to the effective number of cells in the solution. Moreover, to grant the cross-correlation with different experimental runs, the obtained values were normalized to a standard RBC solution with 45% of haematocrit.

### 4.5. Glutathione

The sample’s aliquots (200 µL each) were pre-treated by adding 600 µL of cold 10% metaphosphoric acid and incubated at 4 °C for 15 min, then centrifuged at 20,000× *g* for 15 min at 4 °C. 50 uL and 200 µL of the supernatants were taken for the assay of GSH and GSSG, respectively, and immediately stored at −80 °C.

GSH assay: the 50 µL aliquot of supernatant was mixed with 1.0 mL of 0.1% EDTA in 0.1 M sodium hydrogenphosphate, pH 8.0. Then, an aliquot of 20 µL was taken and treated with 300 µL of 0.1% EDTA in 0.1 M sodium hydrogenphosphate, and 20 µL of 0.1% ortho-phtaldialdehyde (OPA) in methanol. The tubes were well-capped and incubated at 25 °C for 15 min in the dark, the solution was then filtered through a 0.20 μm nylon filter and stored at 4 °C prior to the analysis.

GSSG assay: the 200 µL portion of supernatant was incubated at 25 °C with 200 µL of 40 mM NEM for 25 min in dark environment. Then, 750 μL of 0.1 M NaOH was added and 20 μL of this mixture was taken for measurement of GSSG, using the procedure followed for the GSH assay with the difference that 0.1 M NaOH was employed as diluent instead of EDTA phosphate solution.

Chromatography of GSH and GSSG after their derivatization with OPA was performed at 37 °C using isocratic elution with a HPLC Waters 600 pumps system and an AF Waters Online degasser on a X-Bridge C18, 5 μm, 4.6 × 150 mm column associated with a guard-column of the same material (Waters, Milford, MA, USA). The mobile phase was 15% methanol in 25 mM sodium hydrogenphosphate (*v*/*v*) at pH 6.0. The flow rate was kept constant at 0.5 mL/min. The injection of samples or standards into the column was performed using a Waters 717 plus autosampler in aliquots of 50 µL. The excitation/emission wavelengths were set to 350/420 nm in a Shimadzu RF-551 spectrofluorimetric detector and the instrument control and data acquisition were carried out using the Waters^®^ Millennium^®^ 32 software (Empower 2, 2005). The concentration of GSH and GSSG in the samples was determined from the calibration curve that was performed daily through GSH and GSSG solutions prepared in 1 mM hydrochloric acid and stored at 4 °C until use.

### 4.6. Sample Smears Preparation

The smears used for the AFM and optical characterizations were prepared in duplicate, every 2–3 days starting from a plasma-enriched solution (15 μL of sample plus 15 μL of plasma) that was manually smeared onto a commercial poly-L-lysine coated glass slide (Thermo Scientific, Menzel-Glaser, Waltham, MA, USA) and then air-dried. The addition of plasma ensures a homogeneous dispersion of the cells on the entire glass slide and the quantity used (5 μL for each smear) grant an optimal density for AFM measurements. The plasma was stored at 4 °C but was allowed to reach room temperature prior to its use, in order to avoid thermal stress to the cells. We verified that such smears if properly stored, could last un-modified for years.

### 4.7. AFM Characterizations (Mean Volume, H.R. Imaging and Roughness)

The AFM images were collected using a home-designed microscope at room temperature and in a constant 30% relative humidity. The measurements were performed in contact mode, by using Silicon Nitride Veeco MSCT probes (Veeco Process Equipment Inc., Camarillo, CA, USA) with 0.03 N/m elastic constant, asymmetric pyramidal shape and nominal tip radius of 10 nm and maintaining the maximum force between tip and sample under 1 nN. High resolution images used to measure the membrane’s roughness were collected at a scanning speed of 3–4 s/line and the reproducibility of data was carefully tested. 

The surface roughness of erythrocyte’s plasma membrane has been proved a sensitive parameter to evaluate the structural integrity of the RBCs and, therefore, the mechanical support that the cell skeleton can exert. A detailed discussion of the methodology employed for the measurement is reported in the reference [9], while advantages and peculiarities in the use of membrane roughness has been discussed in [16]. The methodology we used here has been slightly modified compared to the oldest paper on this subject [16] in order to improve, especially in the range of the low values, the determination of the surface roughness and to reduce the measurement error. The present method employs the free software Gwyddion version 2.59 (www.gwyddion.net (released on 4 November 2021)) and consists in selecting several (typically more then 10) sampling areas on the cell membrane, all of fixed, 1 × 1 μm, size. After a background subtraction, all the residual morphological component was removed by fitting X and Y axis with a high-grade polynomial. A 7th grade has been used, although 6th or 8th grades could have been used with similar effectiveness and very limited quantitative differences. After the fitting, the surface roughness was measured using the following formula, where *N* is the number of data point used for the roughness calculation; *Xi* is the height of the *i*-th point and *Xm* is the mean height value.
$$Rrms=\frac{1}{\left(N-1\right)}*\sqrt{\sum_{1}^{N}{\left(Xi-Xm\right)}^{2}}$$

### 4.8. Shape Evaluation by Optical Microscopy (Altered Shapes, Crenated, Spiculed and Flat vs. Biconcave)

Each smear was mapped using an Olympus IX 70 inverted microscope, by performing extensive mapping at 400× magnification. Subsequently, the obtained images were manually counted and each cell morphotype was assigned to one of two major categories: (i) native, biconcave shaped erythrocytes and (ii) altered, non-biconcave shapes. Among this latter group, three major morphological phenotypes were identified, according to the degree of alteration: crenated (CREN), spiculed (SPIC, echinocytes), and spherical (SPHE, spherocytes). To achieve statically relevant results, more than 1000 cells were counted for each sample.

### 4.9. Rejuvenation Procedure

The rejuvenation procedure (also named revitalization) was performed at selected aging times according to a modified De Venuto protocol [26]. The sample, an entire vial, was centrifuged at 3000 rpm for 12 min, the supernatant was discarded, and the remaining cells were diluted 1:5 in the rejuvenation buffer (10 mM inosine, 10 mM pyruvate, 75 mM sodium phosphate, 23 mM NaCl, 5 mM NaOH, pH 7.4) and stored for 3 h at 37 °C. After two subsequent washing procedures (in standard buffer, 3000 rpm, 12 min each passage) the cell was resuspended in the standard buffer at a final haematocrit of 20%. A 200 ul aliquot of the sample was taken before and after the rejuvenation procedure to measure the ATP content. The rationale of this procedure consists in providing the cells with the raw materials needed to build up their own ATP levels, it means that only the cells with an active metabolic machinery will be able to increase their ATP content, in this way we were capable to obtain an insight in the metabolic integrity of the cells.

### 4.10. Data Analysis and Statistical Evaluation

All the data analyses were performed using the software package “Origin 8”, except for the AFM images that were analysed through the open access software Gwyddion version 2.59 (www.gwyddion.net (released on 4 November 2021)).

The ATP, Soret peak and reducing power graphs presented in the manuscript were averaged over two or three independent experiments 

The AFM images are representative of the general behaviour of more than 10 cells from minimally three independent experiments, which showed all the same behaviour. The morphological analyses were the result of the optical analysis of more than 1000 cells collected from at least three independent experiments. 

In all graphs, the data have been presented with their relative standard deviation which was used for statistical comparison between samples. 

The data are freely available on reasonable request to the authors.

## 5. Conclusions

In this work, we studied the aging pathway of human erythrocytes in the presence or absence of a mechanical stimulation provided by a gentle cell shaking performed through a tilting shaker. The aging pathway has been evaluated based on morphological (cell shape, ultra-morphology, membrane roughness) and biochemical (ATP, cell lysis, GHS, and Hb oxidation) factors monitored at different aging times.

The obtained data clearly showed that mechanical stimulation acts as a powerful accelerator of the aging. It induces a remarkable consumption of cellular resources and makes the cells experience extreme metabolic requirements. 

Furthermore, in order to investigate the involvement of the mechano-transduction system in the aging acceleration, we employed molecular inhibitors of the mechano-transduction as well as chemicals that act by reducing the mechano-sensing. In both cases, we reported clear evidence of an involvement of mechano-transduction factors in the regulation of major effectors determining the speed and severity of the aging pathway. Interestingly, the involvement of mechano-transduction has been evidenced in the absence of extracellular calcium in the aging buffer, thus suggesting that the mechanical stimulus can be processed by RBCs through multiple, possibly biochemical, pathways.

To our knowledge, this is the first attempt to directly understand the mutual influence and cross-correlation between cell aging and mechano-transduction with an approach that can provide a thorough view of the cell structure and metabolism.

## Figures and Tables

**Figure 1 ijms-23-10180-f001:**
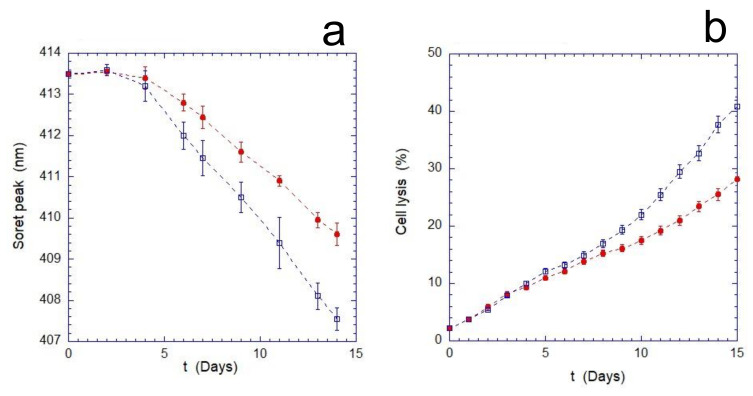
Percentage of cell lysis (**a**) and position of the Soret Peak (**b**) for static (red circles) and shaken (blue empty squares) samples. The overall stress experienced by the mechanically stimulated cells is higher with respect to the static ones at aging times, approximately, longer than 4 days.

**Figure 2 ijms-23-10180-f002:**
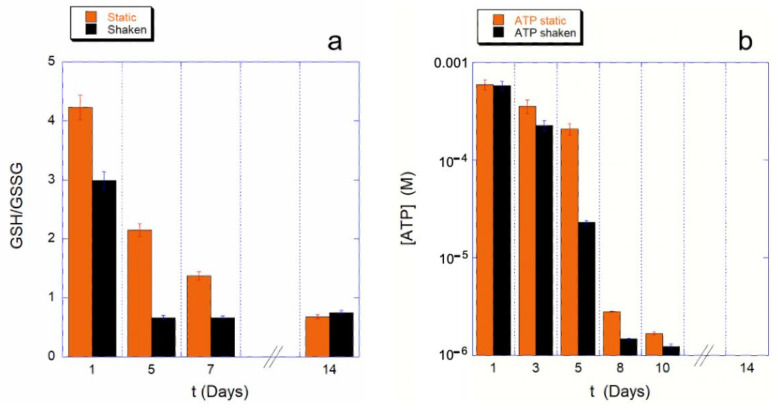
Histograms reporting the GSH/GSGG ratio (**a**) and the intracellular ATP content (**b**) for static (orange bars) and mechanically stimulated (black bars) samples. The energy consumption is higher in the shaken samples with respect to the static ones. In terms of reducing power, this difference is significant already after 1 day of ageing, while in the ATP trend, this difference becomes evident after 4 days of ageing. The ATP concentrations are shown in log scale.

**Figure 3 ijms-23-10180-f003:**
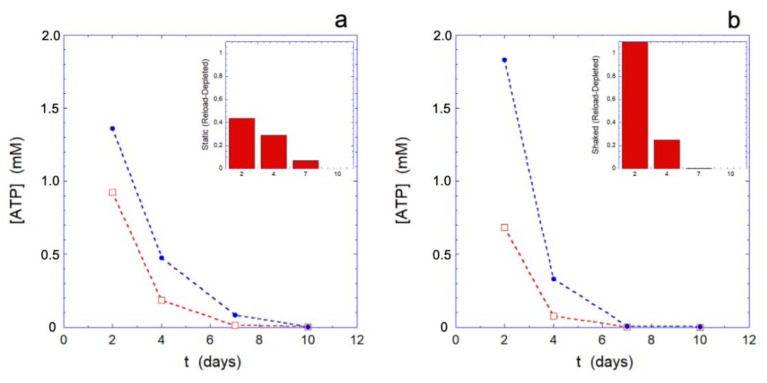
Results of the rejuvenation procedure on static (panel (**a**)) and mechanically stimulated (panel (**b**)) samples. The red squares represent the intracellular ATP content prior to the rejuvenation procedure, while the blue dots represent the intracellular ATP content immediately after the procedure. Insets are shown inside the graphs, to better highlight the ATP increment during the reload. The ATP concentrations are shown in log scale.

**Figure 4 ijms-23-10180-f004:**
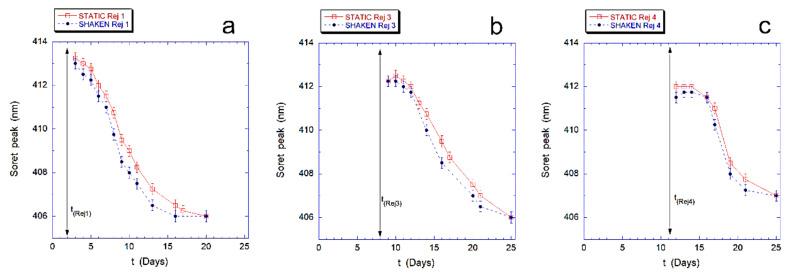
Hemoglobin re-oxidation kinetics after rejuvenation performed at the second (**a**); eighth (**b**), and eleventh day (**c**) for static controls (red squares) and mechanically stimulated (blue circles) samples. The Soret peak in shaken cells always occurs at shorter wavelengths (i.e., is more oxidated) than in static controls. This indicates that Hb re-oxidation is always faster for mechanically stimulated cells, suggesting that the shaken RBCs experience a higher stress condition.

**Figure 5 ijms-23-10180-f005:**
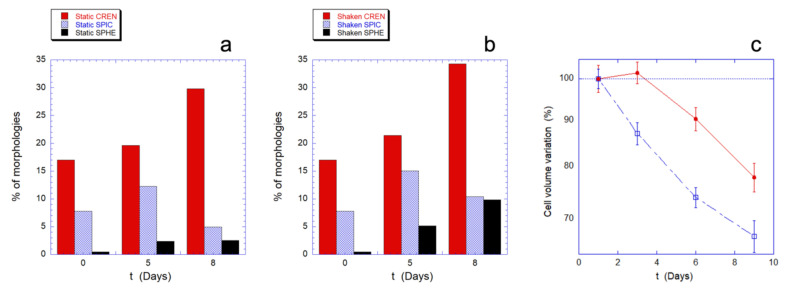
Morphological evolution and volume variation as a function of the RBC aging. Panels (**a**,**b**) show the morphological evolution in absence (**a**) or presence (**b**) of mechanical stimulation. The red, light blue, and black columns represent the percentage of altered, non-biconcave, shapes observed in the samples. The data arise from a count of at least 1000 cells per sample performed with a conventional optical microscope. Therefore, these analyses do not have enough sensitivity on the nanoscale to appreciate the details of the cell membrane but these data allow the analysis of a statistically significant dataset composed of a high number of cells. Panel (**c**) shows the decreasing trend of the mean cell volume, as measured using AFM imaging, for static controls (red circle) and mechanically stimulated cells (open square) as function of the aging. It is worth noting that, for experimental reasons, the optical and the AFM images were collected on alternating days, in such a way that the comparison between these techniques must be performed in term of the overall observed trends.

**Figure 6 ijms-23-10180-f006:**
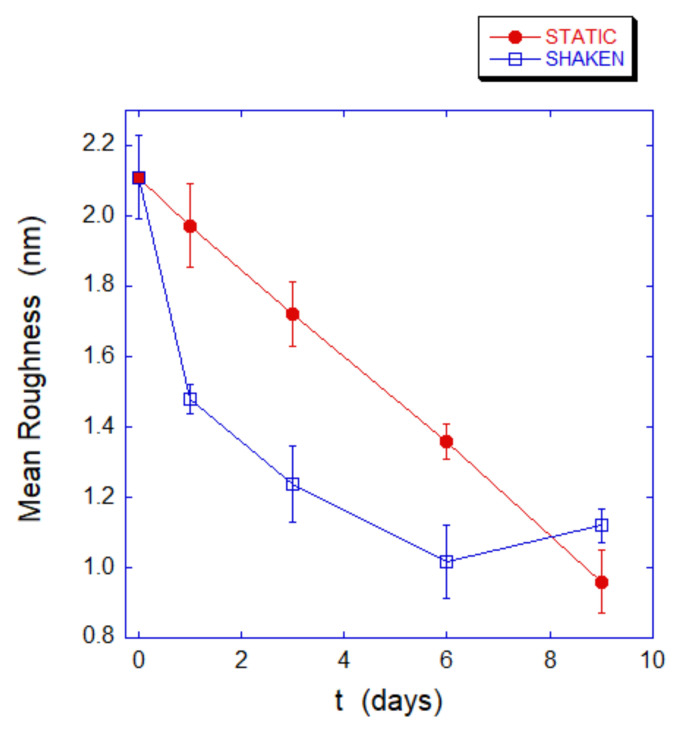
Roughness trend measured for static controls (red circles) and for mechanically stimulated sample (blue squares). The faster decreasing trend in shaken cells indicates a larger disengagement of the cell skeleton from the overlying membrane over time.

**Figure 7 ijms-23-10180-f007:**
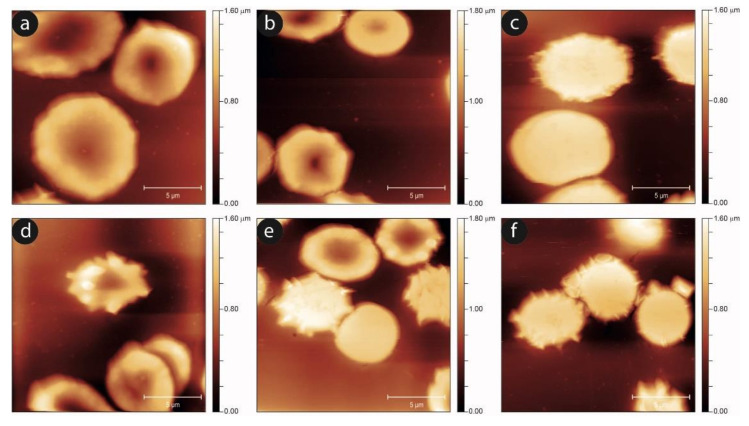
Typical AFM images of the RBCs’ morphology for static controls (upper panels) and mechanically stimulated samples (lower panels) after three (**a**,**d**), six (**b**,**e**), and nine days (**c**,**f**) of aging. Altered morphologies can be observed more often, especially at the longer times, in mechanically stimulated samples than in static controls.

**Figure 8 ijms-23-10180-f008:**
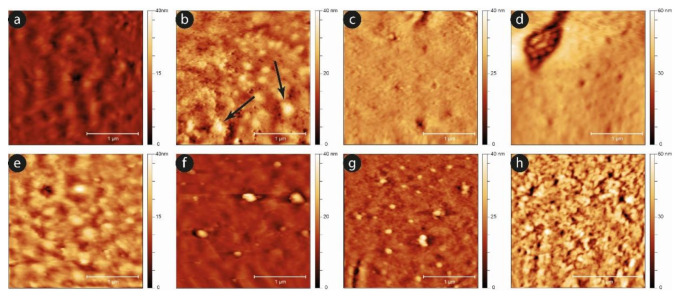
Typical high resolution images of the plasma membrane for static controls (upper panels) and mechanically stimulated samples (lower panels) after one (**a**,**e**), three (**b**,**f**), six (**c**,**g**), and nine days (**d**,**h**) of aging. The data suggest that the occurrence of features such as micro-vesicles and other membrane markers of aging arises sooner in the mechanically-stimulated samples.

**Figure 9 ijms-23-10180-f009:**
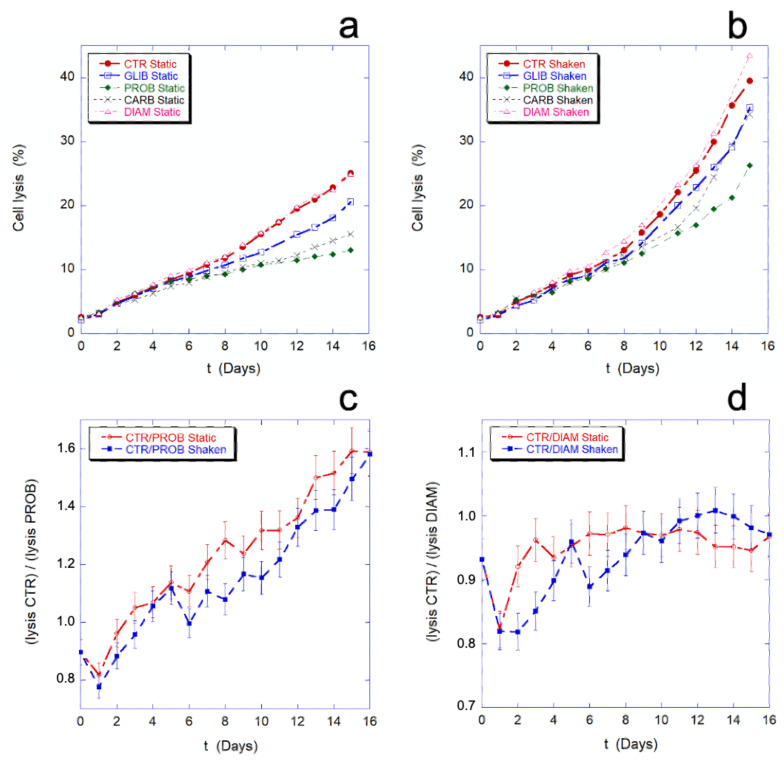
Panels (**a**,**b**) report, respectively, the percentage of cell lysis in static controls (**a**) and in mechanically stimulated cells (**b**) in the presence and absence of DIAM, CARB, GLIB, and PROB. Moreover, in panels (**c**,**d**) we compared the ratio of the lysis curves (shown in Figure 2a,b) for CTR/PROB (**c**) and for CTR/DIAM (**d**). The ratio has been performed both in static and in shaken conditions and is reported in the Y axis of (**c**,**d**). If the mechanical stimulus would provide a specific increase of cell lysis, this would result in systematic differences in the shaken vs. static trends. On contrary, the data show that mechanical stimulus always determines an increase in cell lysis (**a** vs. **b**) but this increasing is not mechanical stimulus-dependent as the two trends in the presence or absence of mechanical stimulus are equivalent (**c**,**d**).

**Figure 10 ijms-23-10180-f010:**
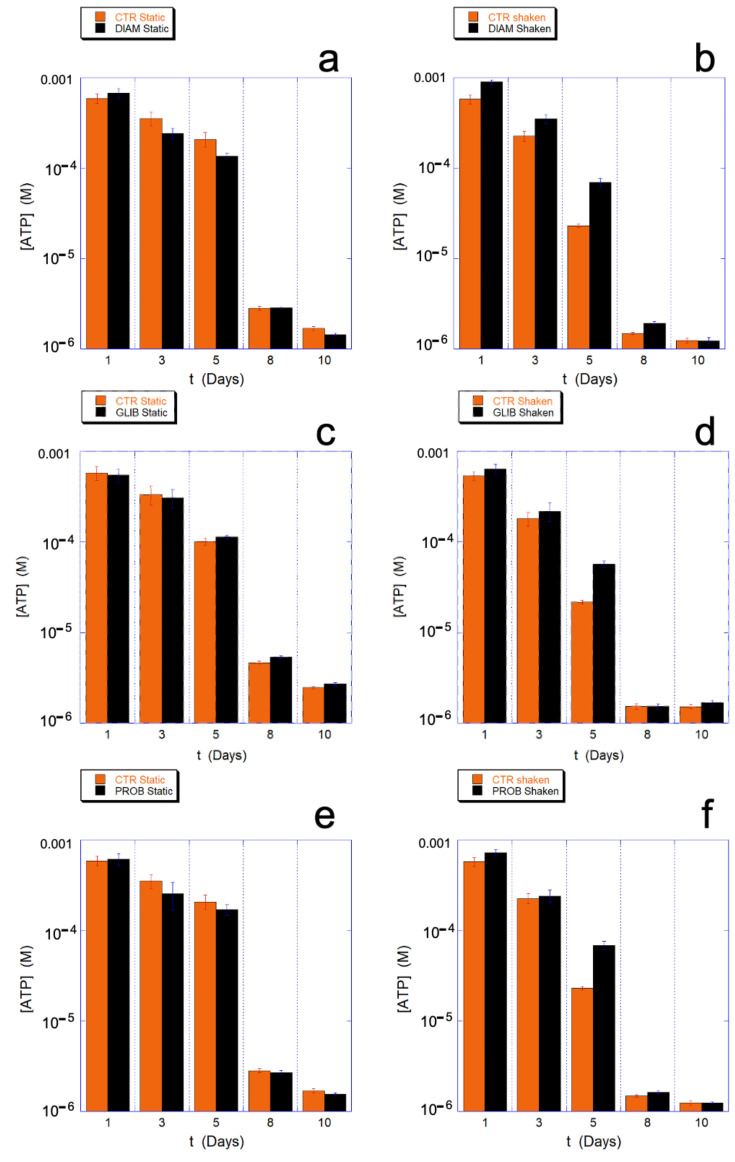
Intracellular ATP measured at various aging times in static (**a**,**c**,**e**) and mechanically stimulated (**b**,**d**,**f**) samples in the presence and absence of drugs that interfere with the mechano-transduction. Respectively, DIAM (**a**,**b**); GLIB (**c**,**d**), and PROB (**e**,**f**) and their specific controls are shown. The data evidence that the drugs have no significant effect in static conditions, while, in the presence of mechanical stimulation, n higher amount of ATP can be measured, at least during the critical aging phase when higher consumption of resource is required (i.e., the fifth day). Data analogous to PROB were measured for CARB. The ATP concentrations are shown in log scale.

## Data Availability

All the data presented in this study is freely available from the corresponding author upon reasonable request.

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
