# Peer review of "Mechano-Transduction Boosts the Aging Effects in Human Erythrocytes Submitted to Mechanical Stimulation"

_ijms, 2022, doi:10.3390/ijms231710180_

Round 1
Reviewer 1 Report
Dinarelli et al. have done several experiments trying to elucidate the erythrocyte aging induced by mechanical stimulation. This paper is well-designed, logically very clear, and very interesting, but the following points could be improved for publication in the IJMS
Major point
Please do a statistical study since there is no statistical analysis.
Minor point
The English expression is difficult to understand. Please proofread it in English.
Author Response
REFEREE 1
Dinarelli et al. have done several experiments trying to elucidate the erythrocyte aging induced by mechanical stimulation. This paper is well-designed, logically very clear, and very interesting, but the following points could be improved for publication in the IJMS
Major point
Q: Please do a statistical study since there is no statistical analysis.
A: We added the information on the statistical analysis in the revised version of the manuscript.
Minor point
Q: The English expression is difficult to understand. Please proofread it in English.
A: In agreement with the referee’s suggestion, we have reread critically the entire manuscript and improved the English accordingly.
Reviewer 2 Report
The manuscript described the study of the erythrocyte aging pathway under static and mechanically stimulated situations. The mechanical stirring was found to accelerate the aging pathway evaluated by morphological properties such as cell shape, ultra-morphology, and membrane roughness, and biochemical factors such as ATP, cell lysis, GHS, and Hb oxidation at different times. The Hb oxidation state was suggested to be involved in the mediation of the metabolic shift between two pathways. However, the mechanical stimulation didn’t affect the cell lysis though the reduction of ATP consumption was related to the regulation of metabolism. The merit of this study is to provide an approach for the study of erythrocyte aging between two pathways in morphologically and metabolically ways. It is recommended for publication in the IJMS after the manuscript is improved according to the following suggestions.
Major:
1. The title may need to be modified to tell what “a role” is. It should be clearly defined.
2. The abstract needs to be improved to reflect the significant results of this study though words are limited. What is the result of this study? What kind of mechanical stimulation was used? What are two pathways? What kinds of drugs and specific inhibitors were used? What kinds of metabolic and morphological factors were investigated?
Minor:
1. Figure 1. The resolution of the image needs to be improved.
2. Figure 2. It is suggested to show the same t (days) for two subfigures. Figure 2a is recommended because the results of intracellular ATP content on days 2, 4, 6, 7, and 9 were not shown.
3. Lines 199-201. It was stated that the re-oxidation kinetics of intracellular Hb that occurs after revitalizations were performed after 2, 4, 8, and 11 days. However, only the results after 2 and 8 days were shown in Figure 4. It is suggested to add the results after 4 and 11 days.
4. Figure 4. The definitions of Rej1 and Rej3 should be described. How do define the meaning of “faster” which was described in the legend in a scientific way?
5. Lines 217-219 and Figure 5. It was described that the cells have been divided into four morphological groups: biconcave, crenated, spiculated, and spherocytic. However, the biconcave and spherocytic shapes were not shown in Figure 5a and Figure 5b but the flat shape. It is suggested to add the essential description of flat shape. In addition, it seems the y-axis of Figure 5c was incorrect. The results of cell volume variation (%) were shown on t (days) 1, 3, 6, and 9 which were inconsistent with those shown in Figure 5a and Figure 5b. Maybe it was a drawing mistake. If the suggestion is right, the authors need to add the result on t (days) 2 to Figure 5a and Figure 5b for comparison in addition to t (days) 0, 5, and 8. Moreover, the inconsistency was also observed in Section 4.8.
6. The position of Figure 6 should be placed after line 260.
7. The mean roughness of the cells was slightly increased at longer aging times after t (days) 8. The authors explained it is due to the loose bounding of the membrane to the skeleton that takes place during the long aging time. It is suggested to add the proofs to demonstrate this speculation.
8. The methods for performing the effects of the employed chemicals and inhibitors on the percentage of cell lysis were not found in the Materials and Methods.
9. Figure 9. The abbreviations of the drugs were inconsistent. It is suggested to show them as four-letter abbreviations in upper case, the same as those shown in the text. In a similar situation, sta and Stat, and shaken and SHAKEN were mixed used which may cause confusion. In addition, the definitions of the y-axis, (CTR/PROB)sta/(CTR/PROB)bas, were not explained.
10. The position of Figure 9 should be placed after line 324.
11. In the same situation as described above, the abbreviations of the drugs should be consistent in Figure 10.
12. Figure 10. The results of intracellular ATP measured for CTR in static and mechanically stimulated samples were different. For example, the intracellular ATP concentration of CTR static was different in Figure 10c from those in Figure 10a and Figure 10e. Similarly, the intracellular ATP concentration of CTR shaken was different in Figure 10d from those in Figure 10b and Figure 10f. How to explain it?
13. The position of Figure 10 should be placed after line 339.
14. Lines 502-503. The directions of the mechanical signal processed through different pathways with the common result of regulating the RBCs metabolism under stress conditions can be further discussed.
Author Response
REFEREE 2
The manuscript described the study of the erythrocyte aging pathway under static and mechanically stimulated situations. The mechanical stirring was found to accelerate the aging pathway evaluated by morphological properties such as cell shape, ultra-morphology, and membrane roughness, and biochemical factors such as ATP, cell lysis, GHS, and Hb oxidation at different times. The Hb oxidation state was suggested to be involved in the mediation of the metabolic shift between two pathways. However, the mechanical stimulation didn’t affect the cell lysis though the reduction of ATP consumption was related to the regulation of metabolism. The merit of this study is to provide an approach for the study of erythrocyte aging between two pathways in morphologically and metabolically ways. It is recommended for publication in the IJMS after the manuscript is improved according to the following suggestions.
Major:
Q1. The title may need to be modified to tell what “a role” is. It should be clearly defined.
A1: According to the referee’s suggestion we changed the title as follow: “Mechano-transduction boost the aging effects for human erythrocytes in the presence of mechanical stimulation”
Q2. The abstract needs to be improved to reflect the significant results of this study though words are limited. What is the result of this study? What kind of mechanical stimulation was used? What are two pathways? What kinds of drugs and specific inhibitors were used? What kinds of metabolic and morphological factors were investigated?
A: Following the suggestions of the referee we have improved and clarified the abstract
Minor:
Q1. Figure 1. The resolution of the image needs to be improved.
A1: we have improved the quality of figure 1
Q2. Figure 2. It is suggested to show the same t (days) for two subfigures. Figure 2a is recommended because the results of intracellular ATP content on days 2, 4, 6, 7, and 9 were not shown.
A2: In agreement with the referee’s suggestion, we have changed the X-axis in order to have both graphs plotted on the same time scale.
Q3. Lines 199-201. It was stated that the re-oxidation kinetics of intracellular Hb that occurs after revitalizations were performed after 2, 4, 8, and 11 days. However, only the results after 2 and 8 days were shown in Figure 4. It is suggested to add the results after 4 and 11 days.
A3: We agree with the referee but, for the sake of clarity, we included in the figure only the results after 2,8 and 11 days, because the data of rejuvenation after 4 days do not add much information as they are substantially identical to the ones at shorter times.
Q4. Figure 4. The definitions of Rej1 and Rej3 should be described. How do define the meaning of “faster” which was described in the legend in a scientific way?
A4: Rej1 is the first rejuvenation process (described in M&M and elsewhere in the text) which was performed after 2 days, while Rej3 refers to the procedure performed after 8 days. In the present version we added the data for the forth rejuvenation procedure (Rej4) which was performed after 11 days of aging.
We have changed the manuscript to clarify the meaning of “faster” in Figure 4. The text now states that: “The Soret peak in shaken cells typically occurs at shorter wavelengths (i.e. Fe-Hb is more oxidated) than in static controls. This indicates that Hb re-oxidation is faster for mechanically stimulated cells, suggesting that the shaken RBCs experience a higher stress condition.”
Q5. Lines 217-219 and Figure 5. It was described that the cells have been divided into four morphological groups: biconcave, crenated, spiculated, and spherocytic. However, the biconcave and spherocytic shapes were not shown in Figure 5a and Figure 5b but the flat shape. It is suggested to add the essential description of flat shape. In addition, it seems the y-axis of Figure 5c was incorrect. The results of cell volume variation (%) were shown on t (days) 1, 3, 6, and 9 which were inconsistent with those shown in Figure 5a and Figure 5b. Maybe it was a drawing mistake. If the suggestion is right, the authors need to add the result on t (days) 2 to Figure 5a and Figure 5b for comparison in addition to t (days) 0, 5, and 8. Moreover, the inconsistency was also observed in Section 4.8.
A5: The biconcave shape was not presented in the graphs in figure 5 since this figure is referred to the modified, altered shapes of erythrocytes, we clarified this aspect in the revised version of the manuscript.
Flat and spherocytic shape are the very same cells, simply called in two different ways, thus we amended the manuscript by substituting the name of the spherocytes to SPHE instead of flat.
The Y-axis of figure 5c is correct, since it represents the percentage of cellular volume with respect to the 100% of volume measured after 1 day of aging (since it is the same as t0) and, according to our AFM measurements, it decreases over time. Maybe the misunderstanding on the days arises from the fact that the graphs 5a and 5b represent optical characterization while 5c arises from an AFM data analysis (we further clarified this aspect in the text). The smears on which the optical manual counting was carried out were not the same used for the AFM analysis, but the trends are consistent.
The inconsistence on section 4.8, we believe, arises from the fact that we did not state clearly that flat and spherocytic cells are the same morphology, we clarified this aspect in the revised version of the manuscript.
Q6. The position of Figure 6 should be placed after line 260.
A6: We shifted the figure to line 260
Q7. The mean roughness of the cells was slightly increased at longer aging times after t (days) 8. The authors explained it is due to the loose bounding of the membrane to the skeleton that takes place during the long aging time. It is suggested to add the proofs to demonstrate this speculation.
A7: Despite the fact that the weakening of the skeleton-to-membrane contact at long aging time is a well-established fact, we changed the sentence according to the referee suggestion. Now the involvement of the membrane rearrangement and ruffling in the small roughness increase is presented more as a likely possibility rather than an obvious consequence of the biophysical cell conditions
Q8. The methods for performing the effects of the employed chemicals and inhibitors on the percentage of cell lysis were not found in the Materials and Methods.
A8: All the requested information was already present in the manuscript. The method to measure the percentage of cell lysis is described, as well as the procedure we used to add the inhibitors to the RBC solutions. There are no differences in the methods or the data analysis when performing the lysis measurements on control or treated cells.
Q9. Figure 9. The abbreviations of the drugs were inconsistent. It is suggested to show them as four-letter abbreviations in upper case, the same as those shown in the text. In a similar situation, sta and Stat, and shaken and SHAKEN were mixed used which may cause confusion. In addition, the definitions of the y-axis, (CTR/PROB)sta/(CTR/PROB)bas, were not explained.
A9: We agree with the referee. We have carefully reviewed the manuscript in order to avoid inconsistencies in the naming.
Now we use 4 letter abbreviation for the drugs all over the text. On contrary, we maintain the use of Static and Shaken, as in our opinion using the full English word for the treatment might help in avoid misunderstanding
Regarding figure 9, we have modified the definition of the Y-axis and clarified its meaning in the text.
Q10. The position of Figure 9 should be placed after line 324.
A10: According to the referee suggestion, we shifted figure 9 after line 324
Q11. In the same situation as described above, the abbreviations of the drugs should be consistent in Figure 10.
A11: We agree with the referee and corrected this point in the revised manuscript.
Q12. Figure 10. The results of intracellular ATP measured for CTR in static and mechanically stimulated samples were different. For example, the intracellular ATP concentration of CTR static was different in Figure 10c from those in Figure 10a and Figure 10e. Similarly, the intracellular ATP concentration of CTR shaken was different in Figure 10d from those in Figure 10b and Figure 10f. How to explain it?
A12: The referee is perfectly right. Indeed, the data reported for glibenclamide has not been measured simultaneously to the other drugs because this drug was part of a secondary experimental plan that involved two distinct glibenclamide experimental run. As consequence, the control file plotted in Glibenclamide graph IS the right control for GLIB, but is not the very same control file used for the other drugs.
Yet, due to this unusual lack of simultaneity, during the glibenclamide experiments we decided to double check the data by using a double control. In the present version of the manuscript the two controls have been averaged (separate files are, clearly, available for referee check) and for this reason slight differences can still be observed compared to the previous version of figure 10.
Obviously, this re-evaluation of glibenclamide data (in fact, of its control) produces no changes in the significance and relevance of the data, but the present procedure is, probably, more solid.
Q13. The position of Figure 10 should be placed after line 339.
A13: In agreement with the referee’s suggestion, the position of Figure 10 is now after line 339
Q14. Lines 502-503. The directions of the mechanical signal processed through different pathways with the common result of regulating the RBCs metabolism under stress conditions can be further discussed.
A14: We tried our best to clarify the relative sentences taking into account our present data. In general terms, the flux of mechanical stimulus in shaken samples has been considered and discussed in the text, also in comparison with the intriguing case of the artificial microgravity stimulation. Clearly, dedicated experiments would be required to completely understand the delicate biochemical and biophysical cellular balance in the presence of complex environmental stimuli.